# The effect of ice shelf rheology on shelf edge bending

W. Roger Buck

Lamont-Doherty Earth Observatory of Columbia University, 61 Rt. 9w Palisades, NY 10960, USA

*Correspondence to*: W. Roger Buck (buck@ldeo.columbia.edu)

**Abstract.** The distribution of pressure on the vertical seaward front of an ice shelf has been shown to cause downward bending of the shelf if the ice is assumed to have vertically uniform viscosity. Satellite lidar observations show that many shelf edges bend upward and that the amplitude of upward deflections depends systematically on ice shelf thickness. A simple analysis is presented showing that upward bending of shelf edges can result from vertical variations in ice viscosity that are an expected consequence of the temperature dependence of ice viscosity and observed ice shelf temperature profiles. Resultant vertical

variations in horizontal stress produce an internal bending moment that can counter the bending moment due to the shelf-front water pressure. Assuming a linear profile of ice temperature with depth and an Arrhenius relation between temperature and strain rate allows derivation of an analytic expression for internal bending moments as a function of shelf surface temperature, shelf thickness and ice rheologic parameters. The effect of a power-law relation between stress difference and strain rate is also included analytically. The key ice rheologic parameter affecting shelf edge bending is the ratio of the activation energy,

$Q$, and the power-law exponent, $n$. For cold ice surface temperatures and large values of $Q/n$, upward bending is expected, while for warm surface temperatures and small values of $Q/n$ downward bending is expected. The amplitude of bending should scale with the ice shelf thickness to the power 3/2 and this is approximately consistent with a recent analysis of shelf edge deflections for the Ross Ice Shelf. These scaling relations should help guide fully two-dimensional numerical simulations of shelf bending.

## 1 Introduction

Ice shelf breakup is important since it could reduce the buttressing of ice sheets, leading to a speed up of ice sheet flow and therefor a rise of sea level (Scambos et al., 2004; Rignot et al., 2004; Schoof, 2007; Gudmundsson, 2013; Fürst et al., 2016). Long-term models of ice sheet flow predict accelerated sea-level rise caused by loss of ice shelves (e.g. DeConto et al., 2021). Ice shelf bending may lead to calving and break up of shelves in two ways. Bending stresses may lead directly to crevasse

formation and propagation (e.g. Wagner et al., 2016). Bending may also affect the routing and pooling of surface meltwater, which can facilitate crevasse growth and calving (e.g. Weertman, 1973; Lai et al., 2020; Buck, 2023).

The rheology of ice is critical to ice-shelf calving and to flow of ice sheets and shelves (e.g. Cuffey and Patterson, 2010). Laboratory and theoretical analyses suggest that ice flow can be described as a non-Newtonian viscous fluid (Glen, 1955) with a strong temperature dependence (e.g. Weertman, 1983). However, there is great uncertainty in the parameters that describe

ice flow (e.g. Cuffey and Patterson, 2010; Behn et al., 2021, Zeitz et al., 2020; Millstein, et al., 2022). Analysis of ice shelf bending may provide an additional constraint on ice rheology.

The downward bending of ice-shelf edges is expected to result from the bending moment due to the pressure in water exerted on the shelf. Weertman (1957) derived an expression for this bending moment as a function of ice and water densities, assuming a uniform ice rheology with depth. Reeh (1968) numerically calculated the deflections due to this bending moment
by treating the ice shelf as a uniform, thin, viscous plate and showed how the downward deflections of the edge would increase with time since the last calving event, as viscous stresses relax. Fully two-dimensional viscous (Mosbeux et al, 2021) and visco-elastic models (Christmann et al, 2019) of ice shelf bending, assuming uniform properties with depth, confirm the thin-plate predictions.

The down-bending of shelf edges predicted by the Weertman-Reeh theory has been seen in several locations such as the edge
of the Ronne Ice Shelf and across several iceberg edges (Scambos et al, 2005 and see Fig. 1(b)). However, upward bending has been observed in several other areas (e.g. Scambos et al, 2005, 2008, Becker et al., 2021: and see Fig. 1(b)). That many shelf edges bend upward with a high "rampart" at the shelf edge paired with a pronounced low "moat" inward of the edge and has generated great interest (e.g. Wagner et al., 2014, 2016; Mosbeux et al, 2021). For example, a comprehensive study of the Ross Ice Shelf by Becker et al (2021) using ICESat-2 lidar shows that moats and ramparts comprise 74% of useable satellite
profiles. Those authors note that the moats and ramparts are separated by horizontal offsets of a few hundred meters and have typical elevation differences of 5 to 10 meters (Fig. 1(c)).

The only published model to explain rampart-moat structures relates to erosion of part of the sub-aerial shelf by wave action (Scambos et al., 2005; Wagner et al., 2014; Mosbeux et al., 2020, Becker et al., 2021). The submarine remnant, termed a "foot" or "bench," then acts as a load that pushes up the un-eroded shelf edge (Fig. 2). Bench-driven up-bending can increase the
magnitude of extensional stresses under the moat that could drive basal crevassing and calving (Wagner et al., 2014; 2016).

This study considers an alternative model to the submerged bench model that depends on vertical variations in the viscosity of an ice shelf. In some sense this work is a continuation of the analysis of Reeh (1968) in that it deals with bending moments causing an ice shelf to flex. In the pioneering paper on that topic, Reeh (1968) wrote: "... a correct treatment of the problem would require consideration of the great variation (by a factor ten or more) of the viscosity. This, however, would involve
enormous mathematical troubles."

Here I show that, as long as we can assume that the viscosity in an ice shelf varies exponentially with depth, the "mathematical troubles" are minimal. The exponential viscosity approximation is shown to be reasonable if the temperature dependence of viscosity can be described by an Arrhenius relation and the temperatures increase linearly with depth in an ice shelf. That approximation allows derivation of scaling relations between ice rheologic parameters, ice surface temperatures and shelf edge

deflections. For "great variations of viscosity" across an ice shelf I show that the edge of the shelf should bend upward to make
a rampart with a corresponding inboard moat. After that I consider the effect of non-linear temperature profiles on shelf
bending. Before launching into a detailed analysis of this problem I discuss some basic ideas about bending moments and layer
bending.

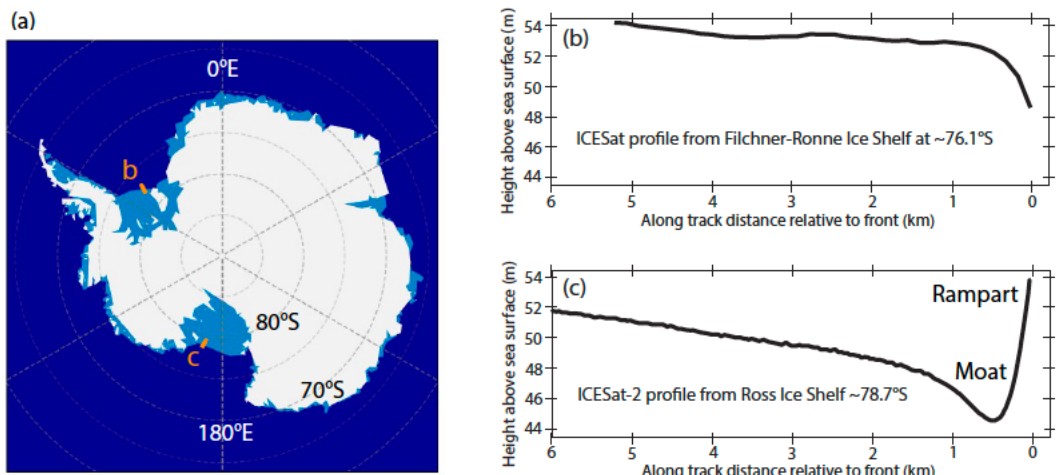

**Figure 1: Locations of topographic profiles and borehole temperature measurements.** (a) Shaded relief map of Antarctica
showing approximate locations of the profiles shown in (b) and (c). (b) ICESat-derived profile of elevations across the edge
of the Ronne Ice Shelf from Scambos et al (2005) showing down-bending of the shelf edge. Profile is oriented 63° relative to
shelf edge. (c) Shelf-edge topography averaged over 4 ICESat-2 lines on the Ross Ice Shelf showing characteristic "moat and
rampart" relief from Becker et al. (2021).

**2 Conceptual model**

Ice shelves that are not heavily buttressed are under extension (Weertman, 1957). While ice shelves are typically assumed to
have negligible vertical gradients in horizontal strain rates, significant vertical variation in viscosity generates vertical gradients
in horizontal stress that cannot be neglected. The idea that stresses internal to a layer can cause it to bend is well known in
engineering and can be illustrated with a bi-metallic strip that bends as temperatures change. Such a strip consists of two metal
layers with different thermal expansion coefficients that are welded together. At a certain temperature this layered strip can be
flat, but it will "curl up" when the temperature is increased as long as the lower layer has a larger thermal expansion coefficient.
This occurs because the lower layer expands more than the upper layer producing vertical variations in horizontal stress.

Internal stresses have been considered in explaining some lithospheric bending observations. Parmentier and Haxby (1986)
showed how vertical variations of horizontal stress within a strong layer might explain downward bending of lithosphere at

transform faults. In this case, the stresses arise due to greater rates of thermal contraction of deeper lithosphere combined with yielding of shallow lithosphere. Those authors note that gravity prevents lithospheric bending that is of much longer wavelength then the effective flexural wavelength of the layer. At very long length scales gravity prevents the layer from bending. The vertical variation in stress can be thought of as an internal bending moment that is everywhere matched by adjacent bending moments except at the plate edge (transform fault) where stresses, and so the applied bending moment, are

different.

The concept of internal bending moments was also applied to the formation axial relief at plate spreading centers. This was first suggested to explain "axial highs" that mark the plate boundary at most fast-spreading centers and typically rise 300-500 meters above the surrounding seafloor. In that case, the lithosphere is accreted with intrinsic curvature that flattens as the plate moves away from the spreading center, as suggested by (Buck, 2001). That paper noted that the deflection of a plate with a

free end and a uniform internal moment are equivalent to that resulting from application of an opposite moment to the end of a layer with no internal moment. Axial valley lithosphere is accreted with the opposite sense of curvature as axial high lithosphere (e.g. Liu and Buck, 2018) and recently observed reverse fault earthquakes close to axial valleys is attributed to the flattening of these curved plates plates with distance from the spreading axis (Olive et al., 2024).

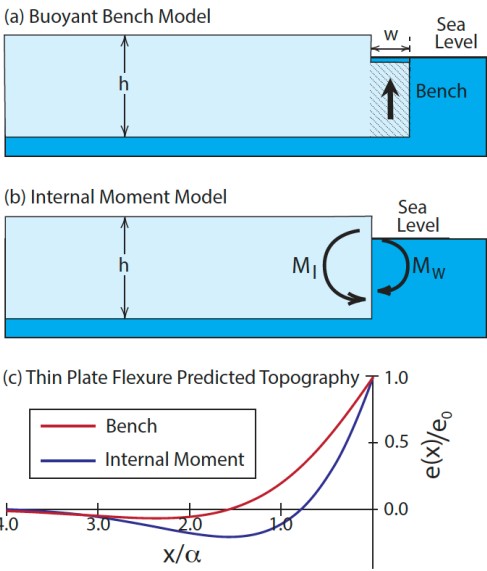

**Figure 2: Illustrations of the bench and internal moment models for deflection of a shelf edge**. The arrow on (a) indicates the upward force due to a submerged ice "bench." The internal moment arises from vertical variations in horizontal stress indicated by horizontal arrows on (b). $M_W$ and $M_I$ are the applied moments due the pressure distribution of water and the internal stress variations, respectively. (c) shows the thin plate model predictions of the model vertical deflections $e$ as a

function of distance from a shelf edge for Eq. (21) and (23), where $e_0$ is the deflection at the shelf edge (see text for explanation).

A major difference between an ice shelf and the lithosphere (or a bi-metallic strip) is that the stresses in the ice are controlled by viscosity. The viscous, or more precisely the viscoelastic, response of an ice shelf to bending moments was recognized by Reeh (1968) who noted that the wavelength of the response after a calving event should decrease with time. This cannot be properly treated without a fully 2D numerical simulation, but, as discussed at the end of this paper, this effect should lead to slow growth of the amplitude of bending deflections. As noted by Reeh (1968) the top of an ice shelf is typically colder than the base (by as much as 30 °C) and so viscosity decreases with depth in the shelf. There is a corresponding decrease in extensional stress with depth and this affects the internal bending moment in the shelf.

An internal moment will not cause bending where the shelf is laterally uniform and continuous over distances much larger than the flexural wavelength. Imagine that a calving event just broke off a broad section of the shelf making a new shelf edge. If the bending moment applied by air and water at the shelf edge is different from the internal bending moment then the shelf edge should bend, much like the bi-metallic strip described above. However, the bending will take time to develop as viscous flow causes the flexural wavelength to diminish with time.

In the absence of a bench, it is the internal stress distribution that determines whether the edge bends up or down. The key question for this paper is how the horizontal stresses internal to an ice shelf affect the bending of a shelf edge. To address this question, I follow the approach of Weertman (1957) and Reeh (1968) and calculate the contribution to the total applied bending moment due to the difference between water pressure and internal stress in the ice layer. The new twist is that I consider internal stress variations related to vertical viscosity variations in the ice shelf.

## 3 Internal Moment Model

### 3.1 Stresses, Pressures and Bending Moments

To determine how an ice layer will flex due to the water pressure distribution on the side of the layer (as shown in Fig. 3) we need to calculate the total applied bending moment $M_T$ due to the difference between pressure of the water $P_w$ and the horizontal stress $\sigma_{xx}$ in the ice:

$$M_T = -\int_0^h [P_w(z) - \sigma_{xx}(z)]z\,dz \tag{1}$$

where the arbitrary minus sign ensures that upward bending corresponds to positive total applied moments. The pressure in the water that acts on the side of the floating ice layer is:

$$P_w(z) = \begin{cases} 0 & for \ z < d \\ \\ \rho_w g(z - d) & for \ h > z > d \end{cases} \tag{2}$$

where $\rho_w$ is the density of water, $h$ is the thickness of the ice layer, $z$ is depth below the ice surface $g$ is the acceleration of gravity and $d$ is the freeboard height shown on Fig. 3. Equating the vertical stress and water pressure at $z = h$ requires that:

$$d = \left(\frac{\rho_w - \rho_i}{\rho_w}\right) h \tag{3}$$

assuming that the vertical stress, $\sigma_{zz}$, in the ice is:

$$\sigma_{zz}(z) = \rho_i g z \tag{4}$$

where $\rho_i$ is the density of ice. I use the geologic convention that positive stress is compressive to simplify the comparison of water pressures and ice stresses. Horizontal force balance requires that:

$$F_x = \int_0^h \sigma_{xx}(z) dz = \int_d^h P_w(z) dz \tag{5}$$

where $\sigma_{xx}(z)$ is the horizontal stress in the layer.

To consider the effect on layer edge bending for a range of possible distributions of the horizontal stress that satisfy Eq. (5) it is useful to define a reference horizontal stress distribution $\sigma_{xxR}(z)$. This allow separate calculation of the applied moments due to the water pressures, $M_W$, and the horizontal stresses with the ice, $M_I$, relative to that reference, such that Eq. (1) can be re-written:

$$M_T = M_W + M_I = -\int_0^h [P_w(z) - \sigma_{xxR}(z)] z dz + \int_0^h [\sigma_{xx}(z) - \sigma_{xxR}(z)] z dz. \tag{6}$$

The reference horizontal stress distribution is what would obtain for a shelf with uniform rheologic properties, so that the horizontal stress is lower than the vertical stress by a uniform amount, $\Delta\bar{\sigma}$. As noted by Weertman (1957), the value of the average stress difference $\Delta\bar{\sigma}$ at the edge of an ice shelf (or within a shelf where buttressing forces are zero) is constrained by Eqs. (5), (4) and (2) and can be written:

$$\Delta\bar{\sigma} = \bar{\sigma}_{zz} - \bar{\sigma}_{xx} = \rho_i g \frac{h}{2} - \frac{F_x}{h} = \frac{1}{2}\left(\frac{\rho_i}{\rho_w}\right)(\rho_w - \rho_i) g h, \tag{7}$$

so that:

$$\sigma_{xxR}(z) \equiv \sigma_{zz}(z) - \Delta\bar{\sigma} = \rho_i g z - \frac{1}{2}\left(\frac{\rho_i}{\rho_w}\right)(\rho_w - \rho_i)gh. \tag{8}$$

For $\Delta\bar{\sigma}$ given by Eq. 7 the bending moment term related to the water pressure variation is:

$$M_W = -\int_0^h [P_w(z) - \sigma_{xxR}(z)]z\,dz = -\frac{1}{12}\left(\frac{\rho_i}{\rho_w}\right)(\rho_w - \rho_i)gh^3\left[1 - \frac{2d}{h}\right], \tag{9}$$

which is equivalent to that found by Weertman (1957) and Reeh (1968).

The internal bending moment term $M_I$ is:

$$M_I = \int_0^h [\sigma_{xx}(z) - \sigma_{xxR}(z)]z\,dz \tag{10}$$

and below I consider cases for different distributions of the horizontal stress with depth in the ice.

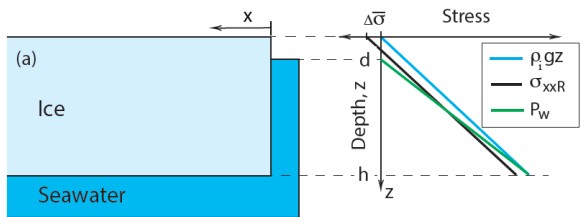

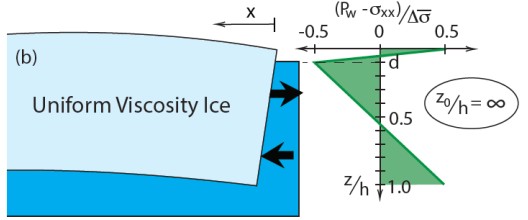

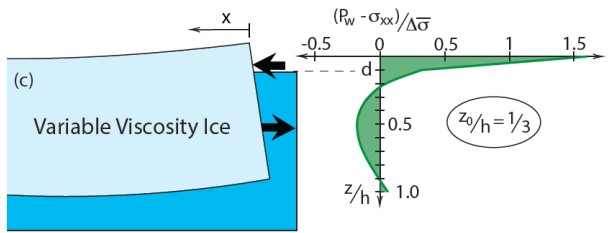

**Figure 3: Illustrations of stresses and stress differences affecting a floating ice layer.** (a) shows an ice layer of thickness $h$ floating on seawater. The sea surface is at a depth $d$ below the top of the ice. The green line shows pressures in seawater and the blue shows the vertical stress in the ice. The black line shows the reference horizontal stress $\sigma_{xxR}$ in the ice which is offset

by a constant stress difference $\Delta\bar{\sigma}$ from the vertical stress. Plot in (b) shows the difference between the water pressure and horizontal stress for a case with uniform properties so that the horizontal stress is the reference stress. (c) shows the difference between the water pressure and horizontal stress for an example with a strong exponential variation of viscosity and so horizontal stress with depth in the ice (with an e-folding depth, $z_0 = h/3$ in Eq. (16)). Cartoons in (b) and (c) show that the sense of bending is the opposite for the cases shown.

The simplest case is for an ice layer with vertically uniform properties and infinite yield strength (or infinite fracture toughness) so there is no opening of surface or basal crevasses, thus reducing the stresses in the layer. This implies a constant offset between the horizontal and vertical stresses in the layer such that $\sigma_{xx}(z) = \sigma_{xxR}(z)$ implying that $M_I = 0$ so that $M_T = M_W$. In this case the moment applied to the end of the layer bends it down. Assuming $\rho_i/\rho_w = 0.9$ in Eq. (3) means that $d = h/10$ so that $M_T = M_W = -\left(\frac{1}{15}\right)\left(\frac{\rho_i}{\rho_w}\right)(\rho_w - \rho_i)gh^3$.

Reeh (1968) calculated that the the applied bending moment increases by up to 30% as the layer is deflected, because as the shelf edge moves down the water pressure on the end increases. However, this neglects the counter effect of the change in pressure on the underside of the deflected layer inboard of the edge. Thus, the average horizontal stress in the ice should remain constant as long as the thickness of the shelf does not change or the top of the layer does not drop below the water surface. Here, I neglect any changes in the applied moment with layer deflection.

## 3.2 Exponential Variation of Effective Viscosity with Depth

As noted above, viscosity in an ice shelf is expected to decrease with depth (e.g. Reeh, 1968). Two simplifying assumptions are used here to relate viscosity variations with depth to rheologic parameters and ice shelf surface temperatures. The first assumption is that temperatures linearly increase with depth. The base of ice shelf should be at the pressure melting point while the surface must be colder (e.g Cuffey and Patterson, 2010) and borehole measurements on some ice shelves indicate nearly linear temperature-depth profiles, as is discussed below.

The second assumption is the form of the ice flow law. There is debate about how the flow of ice varies with stress and temperature and there is evidence that multiple processes require complex descriptions (e.g. Behn et al, 2021). However, a wide range of observations and laboratory data are well approximated with a power-law relation between stress and strain rate, such as Glen's flow law (Glen, 1955), and an Arrhenius relation between strain rate and temperature (e.g. Cuffey and Patterson, 2010). Then the strain rate $\dot{\varepsilon}$ is related to stress difference $\Delta\sigma$ and absolute temperature $T$ as:

$$\dot{\varepsilon} = A\Delta\sigma^n exp\left(\frac{-Q}{RT}\right) \tag{11}$$

where $A$ and $n$ are constants, $Q$ is the activation energy and $R$ is the universal gas constant. The effective viscosity $\eta$ ($\equiv \Delta\sigma/\dot{\varepsilon}$) at a constant strain rate is then:

$$\eta(\dot{\varepsilon}, T) = A^{-1/n} \varepsilon^{\left(\frac{1}{n}-1\right)} exp\left(\frac{Q}{nRT}\right). \tag{12}$$

A constant strain rate is used because the horizontal strain rate should be constant with depth for a uniform thickness ice shelf far from the shelf edge (i.e. where x>>h and x is distance from the edge).

For a constant temperature gradient $dT/dz$ we can describe the temperature with depth in the ice as:

$$T = T_s + \frac{dT}{dz}z = T_s + \frac{(T_B - T_s)}{h}z \tag{13}$$

where $T_S$ is the temperature at the surface and $T_B$ is the temperature at the base of the ice shelf as shown in Fig. 4. Assuming
that $Q/n$ is constant with temperature and that temperature variations in the ice are small compared to the absolute surface temperature, allowing approximation of Eq. (12) as:

$$\eta(\dot{\varepsilon}, z) \cong \eta(\dot{\varepsilon}, T_s)exp\left(\frac{-z}{z_0}\right) \tag{14}$$

where $z_0$ is the "e-folding" length for viscosity variations (i.e. the distance over which the viscosity drops by $1/e$). By ensuring that Eq. (14) a gives the same viscosity as Eq. (12) for the top and bottom of the layer (i.e. where $T(0) = T_S$ and $T(h) = T_B$)
gives:

$$z_0 = \frac{nRT_BT_s}{Q^{dT}/dz} \quad \text{or} \quad \frac{z_0}{h} = \frac{nRT_BT_s}{Q(T_B - T_s)}. \tag{15}$$

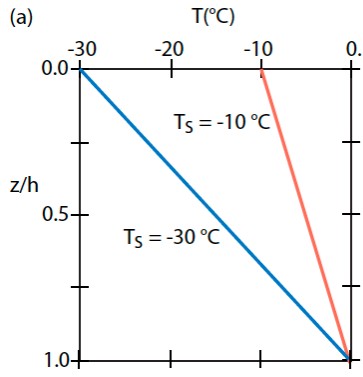

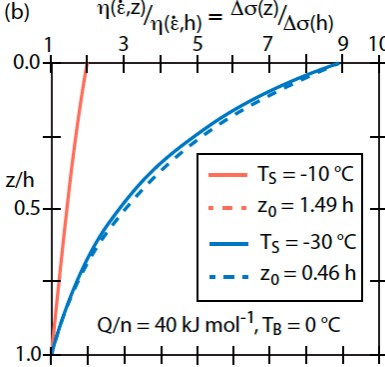

**Figure 4: Illustration of stress differences for full flow flow law and exponential approximation.** (a) Relations between temperatures with depth divided by the ice layer thickness in the ice shelf and (b) the ratio of effective viscosity (or stress difference) at a constant strain rate to the viscosity (or stress difference) at the base of the layer.

The solid lines are based on the standard Glen-type flow law of Eq. (12) and the dashed lines are for the exponential approximation of Eq. (14). For a surface temperature $T_S = -10$ °C there is no visible difference in the two values of viscosity or stress difference and for $T_S = -30$ °C the difference is minor.

For a thin floating layer, the horizontal strain rate far from the sides (i.e. many layer thicknesses) should be constant so that the difference between the horizontal stress and the vertical stress is well approximated as:

$$\Delta\sigma(z) = \sigma_{zz}(z) - \sigma_{xx}(z) = \eta(\dot{\varepsilon}, z)\,\dot{\varepsilon} \cong \Delta\sigma_0 exp\left(\frac{-z}{z_0}\right) \tag{16}$$

where the stress difference at the surface is:

$$\Delta\sigma_0 = \eta(\dot{\varepsilon}, T_S)\,\dot{\varepsilon} = \left(\frac{\dot{\varepsilon}}{A}\right)^{\frac{1}{n}} exp\left(\frac{Q}{nRT_S}\right). \tag{17}$$

The relationship between $z_0/h$ and $T_S$ is shown in figure 5 for a range of values of $Q/n$ assuming for simplicity that $T_B = 0$ °C.

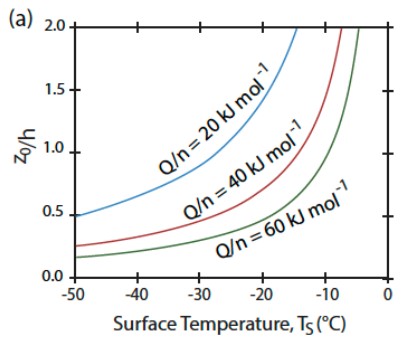

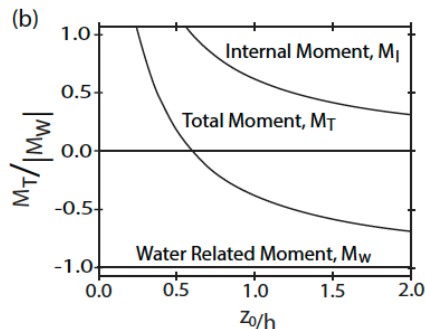

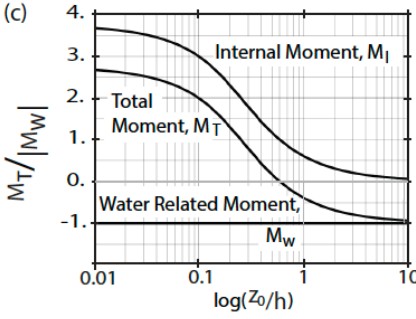

**Figure 5: Relations between model parameters.** (a) Variation of the non-dimensional e-folding depth scale $z_0/h$ with surface temperature assuming the rheologic parameters indicated and that the base of the shelf is at a temperature of 0 °C. (b) and (c) Show the components of the moments as functions of $z_0/h$.

We do not need to know the strain rate to find $\Delta\sigma_0$ since it set by the force applied by water at the front of the ice shelf given
by Eqs (2) and (5). Integration of Eq. (16) over the depth range of the ice shelf implies that:

$$\Delta\sigma_0 = \Delta\bar{\sigma}\left(\frac{h}{z_0}\right)\left[1 - \exp\left(\frac{-h}{z_0}\right)\right]^{-1} \tag{18}$$

Figure 4 confirms that, with a linear temperature profile through an ice shelf, Eq. (14) is an excellent approximation to the Glen's flow law relation given by Eq. (12) for reasonable rheologic parameters and surface temperatures. It shows how the horizontal stress difference varies relative to the stress at the layer base for values of $z_0/h$ defined for given values of $Q/n$ and
225 for 2 values of the surface temperature. For surface temperature warmer than -10 °C the horizontal stresses at all depths are nearly equal to the basal stress and that implies a nearly constant offset between the horizontal and vertical stresses. For colder surface temperatures the horizontal stress differences near the surface are far more extensional than deeper in the layer.

The internal bending moment given by Eq. (10) for the stress distribution of Eq. (18) is:

$$M_I = \frac{1}{2}\left(\frac{\rho_i}{\rho_w}\right)(\rho_w - \rho_i)gh^3\left\{\frac{1}{2} - \left[\frac{z_0}{h} - \left(\frac{z_0}{h} + 1\right)\exp\left(\frac{-h}{z_0}\right)\right]\left[1 - \exp\left(\frac{-h}{z_0}\right)\right]^{-1}\right\} \tag{19}$$

Eq. (19) implies that the internal bending moment and so the bending of a layer depends on the ice and seawater densities, the layer thickness, $h$, and the e-folding depth, $z_0$, for viscosity variations. Figure 5 shows the relation between the internal moment and $z_0/h$. For $z_0 \gg h$ the internal moment is zero which makes sense because the horizontal stresses are equal to the reference stresses at all depths. For $z_0 \ll h$ the maximum value of $M_I$ is $\Delta\bar{\sigma}h^2/2$ which is 3.75 time the absolute value of $M_W$. If $z_0/h \approx$ 0.6 the internal moment is equal and opposite in sign to the moment applied by the water. This would give no bending of the
layer. For larger values of $z_0$ the layer bends down and for smaller values the layer bends up! Figure 6 shows how the surface temperature affects the total bending moment for different values of the rheologic parameter $Q/n$.

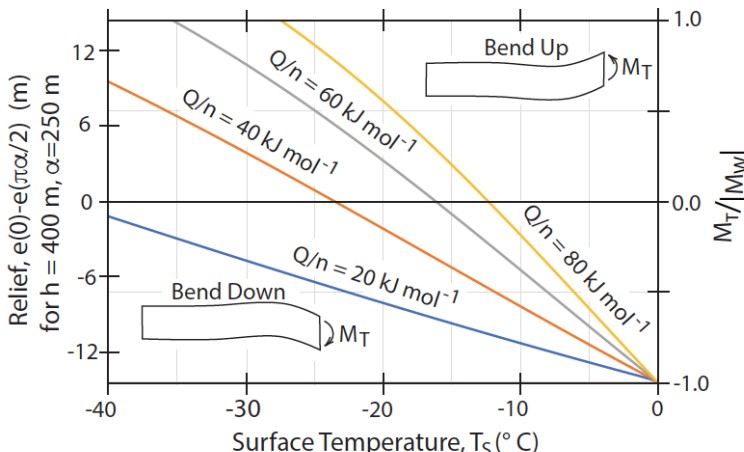

**Figure 6: Analytic model predictions for a range of surface temperatures.** Values of the rheologic parameter $Q/n$ are shown on the different curves. The vertical scale on the right shows the total moment divided by the absolute value of the moment of water pressure for a uniform rheology ice shelf. The scale on the left is the predicted height difference between the shelf edge and the closest position with zero slope (i.e. where the surface is horizontal) for a thin plate with the indicated thickness and flexural parameter.

The error in the analytic approximation was analysed by carrying out numerical integration of the stress differences for the full ice flow law (Eq. (11)). The difference between the full and approximate solutions for the internal bending moment depend on the assumed flow law parameters and surface temperature, but are less than 3% for the most extreme cases illustrated in Fig. 6.

**3.3 Topographic variation based on the thin plate approximation**

To estimate the deflections expected for the bending moments derived here we can use the thin-plate flexure approximations. Many studies of ice shelf bending treat an ice shelf using the plate approximations either with elastic, viscous or viscoelatic rheologies (e.g. Reeh, 1968; MacAyeal and Sergienko, 2013; Olive et al., 2015; Wagner et al, 2016; Banwell et al., 2019; MacAyeal et al., 2021). Vertical deflections of a thin, semi-infinite plate with a moment $M_T$ applied to the end are given in Turcotte and Schubert (2014) as:

$$e^M(x) = e_0^M \exp\left(-\frac{x}{\alpha}\right)\left[\cos\left(\frac{x}{\alpha}\right) - \sin\left(\frac{x}{\alpha}\right)\right]$$

with (20)

$$e_0^M = \frac{2M_T}{\rho_w g \alpha^2}$$

where $x$ is distance from the shelf edge, $\alpha$ is the flexure parameter (directly related to the flexural wavelength), $\rho_w$ is the water density, and $g$ is the acceleration of gravity.

Reeh (1968) and Olive et al. (2016) find that for a viscous or viscoelastic plate with a uniform viscosity $\eta$ the wavelength of the flexure changes with time as:

$\qquad \alpha(t)/\alpha(0) \sim \left[\frac{\tau_M}{t}\right]^{1/4}$

and $\hspace{22em}$ (21)

$\alpha(0) \sim \left[\frac{Eh^3}{\rho_w g(1-v^2)}\right]^{1/4}.$

where $t$ is time, $E$ is Young's Modulus, $v$ is Poisson's ratio and $\tau_M = \frac{E}{\eta}$ is a measure of the Maxwell time of the layer with $\bar{\eta}$ representing the average layer viscosity. Combining Eq. (20) and (21) suggests that the amplitude of deflection should increase

$\quad$ with time roughly as: $e_0^M(t) = e_0^M(0)\left[\frac{t}{\tau_M}\right]^{1/2}$. Eventually, as the flexure parameter approaches the layer thickness, the two-dimensional nature of the problem means that the thin-plate approximation is no longer valid. For layer of a few hundred meters thickness this should take about 1000 Maxwell times. For an average layer viscosity of a few times $10^{14}$ Pas and a layer thickness of 300 m this should take about 4 years. Reeh (1968) came to this conclusion and estimated that the long-term flexure parameter can be a bit smaller than the layer thickness. A more thorough study was done by Olive et al. (2016) who compare

$\quad$ fully two-dimensional viscoelastic models of flexure to the thin plate solution and find the best fitting thin plate flexure parameter evolution to match the 2D results. They find that after many Maxwell times that the effective flexure parameter is smaller than the layer thickness. Thus, for Figs 6 and 7 $\alpha$ is set to 250 m while the ice layer thickness is taken to be 400 m.

Figure 7 shows how the ice surface temperature affects the elevation given by Eq. (20) for a total moment given by the sum of Eqs. (9) and (19). Figure 6 shows just the predicted deflection of the shelf edge for this model. Given that the internal and

275 $\quad$ water-related moment scale with $h^3$ (via Eqs. (9) and (19)) then the combination of this result with Eq. (20) and (21) implies that for the same rheology, temperature profile and time since calving that the deflection amplitude should scale with $h^{3/2}$.

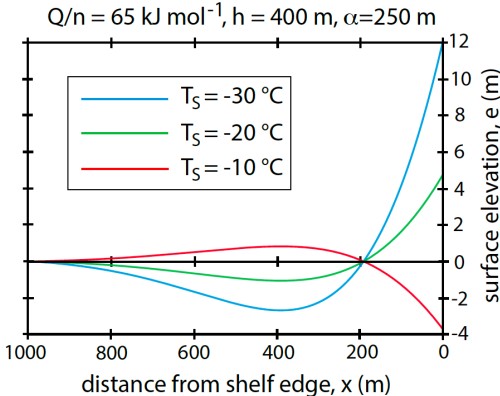

**Figure 7: Internal moment model topography**. Deflections versus distance from a shelf edge are calculated for a thin plate subject to an applied moment at x=0 that depends on the indicated rheologic constant and surface temperatures The model ice shelf is taken to be 400 m thick and the flexure parameter $\alpha$ is set to 250 m.

The thin plate approximations can also be used to illustrate the predictions of the bench model. The effect of the load $V_B$ of a submerged bench on topographic deflections, which (after Wagner et al., 2016) can be written:

$$e^B(x) = e_0^B \exp\left(-\frac{x}{\alpha}\right)\cos\left(\frac{x}{\alpha}\right)$$

(22)

$$e_0^B = \frac{2V_B}{\rho_w g \alpha}$$

where $V_B = w(h-d)(\rho_w - \rho_i)g$ with $w$ being the width of a bench whose top is just at sea level. Combining this relation for the bench load with Eqs. (21) and (22) suggests that the amplitude of vertical deflections for the bench model should scale linearly with the bench width, $w$, but only depend weakly on the ice layer thickness (i.e. bench-driven deflections should scale with $h^{1/4}$). Figure 2 illustrates shows that for the same flexure parameter, $\alpha$, the internal moment model produces a much shorter wavelength response than the bench model.

### 3.4 Effect of nonlinear temperature-depth profiles on shelf bending

Several effects, including accretion or melting of the surface or base of an ice shelf, can contribute to non-linearity of temperatures with depth and this should affect viscosity, stresses and so internal moments. Observations of temperature profiles are limited since they require boreholes through ice shelves and Fig. 8 shows temperature profiles for the three largest Antarctic ice shelves. Two of the three profiles from the Ross Ice Shelf (from Taylor and MacAyeal (1979)) show nearly linear temperatures with depth (a and b on Fig. 8). The profile from location LAV on RIS (Fig. 8 d) shows a concave up profile that

Taylor and MacAyeal (1979) interpret to result from rapid surface accretion. In contrast, the profile from the Amory Ice Shelf (Fig. 8 d) shows a departure from linearity that Craven et al. (2009) assert to result from accretion of marine ice at the shelf base.

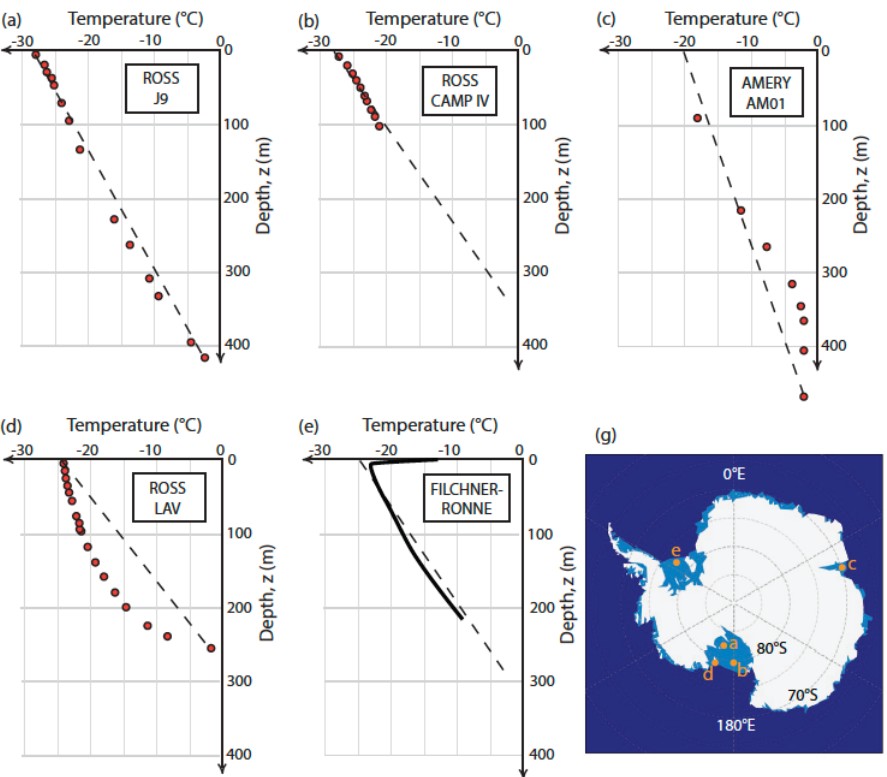


**Figure 8: Borehole temperature measurements for parts of three Antarctic ice shelves.** (a, b, c and e) show relatively linear temperature-depth profiles: (a) and (b) are from open areas of the Ross ice shelf and (c) is from the Amery Ice Shelf. (d) is from an area of the Ross Ice Shelf close to Roosevelt Island. (e) is a profile from the Filchner Ice Shelf redrawn from Eicken et al. (1994). Data points for (a), (b) and (d) are re-plotted from Taylor and MacAyeal (1979), (c) is re-plotted from Craven et al. (2009). Approximate locations of the boreholes in (g).

To estimate the possible generation of such non-linear temperature profiles and their effect on ice-shelf internal bending moments I use a standard ice-shelf thermal model. The approach, described by Robin (1955), assumes that pure shear thinning of the layer maintains a uniform shelf thickness, $h$, while the vertical velocity of the surface is $v_S$ and the vertical velocity of the base is $v_B$. The surface is maintained at $T_S$ and the base at $T_B$. Assuming downward velocity is positive, the steady-state temperature $T$ as a function of depth below the ice surface, $z$, can be written:

$$T(z^*) = T_S + (T_B - T_S) \left\{ \frac{\mathrm{erf}\,(\xi z^*) - \mathrm{erf}\,(-\xi z_{ref})}{\mathrm{erf}\,[\xi(1 - z_{ref})] - \mathrm{erf}\,(-\xi z_{ref})} \right\} \tag{23}$$

where $z^* = z/h - z_{ref}$, $z_{ref} = \frac{v_S}{(v_S - v_B)}$, $\xi = \sqrt{\frac{(v_S - v_B)h}{2\kappa}}$ and $\kappa$ is the thermal diffusivity, here taken to be $10^{-6}$ m²/s. Resulting temperature profiles for a layer 400 m thick and several combinations of $v_S$ and $v_B$ are shown in Fig. 9(a). Freezing onto the base of an ice shelf results in an increase in the temperature gradient with depth while surface accretion results in the opposite

effect on the temperature profile.

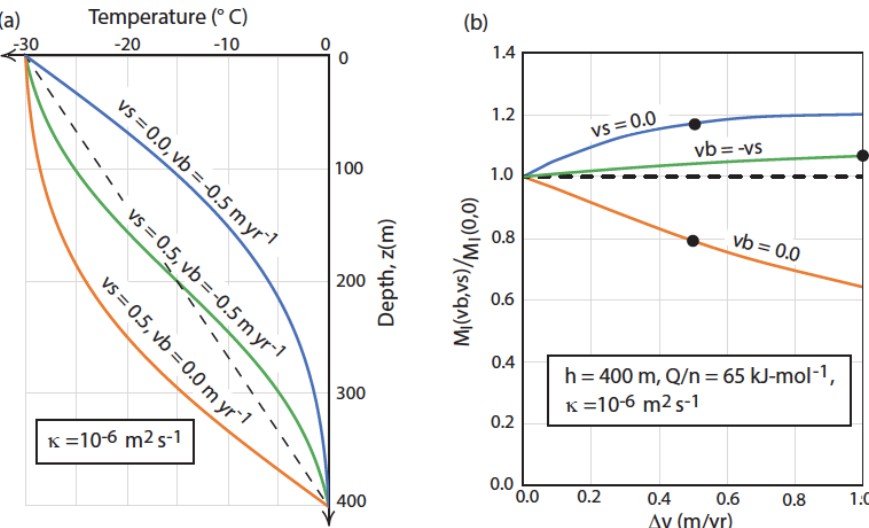

**Figure 9: Effect of surface or basal accretion on ice shelf temperature profiles and internal bending moments.** (a) Examples of three steady-state temperature profiles for the indicated values of surface and basal velocities ($v_S$ and $v_B$) compared with a linear temperature profile for the same surface and base temperatures. (b) Numerically calculated normalized

internal bending moments for a range of temperature profiles calculated with the indicated parameters. Black dots indicate the temperature profiles shown in part (a). Moments are divided by the moment for a linear temperature profile.

Using a temperature depth distribution given by Eq. (23) I use Glen's Flow Law (Eq. (11)) with given rheologic parameters to calculate stress difference variations with depth as a function of strain rate. Then, numerical integration of Eq. (5) is used to set the value of the strain rate in Eq. (11) that gives stress differences that balance the horizontal force applied by water at the

shelf edge. Finally, numerical integration of Eq. (10) with these stress differences gives the internal bending moment ($M_I$).

Figure 9(b) shows examples of fractional variations in the internal bending moment (i.e. the ratio of the internal moment for a given temperature profile divided by the internal moment for a linear temperature profile) for a range of "Robin-type"

temperature profiles for a 400 m thick model ice shelf. The black dots show the fractional variation in $M_I$ for the three particular temperature profiles shown in Fig. 9(a). Freezing onto the base of an ice shelf acts to increase the amplitude of the internal
moment and so the expected upward bending of the shelf edge. In contrast, accretion to the surface and melting of the base act to diminish the internal bending moment. These effects can be large enough to change the sign of shelf-edge bending for very rapid surface accretion or basal melting. Whether this happens depends not only on accretion or melt rates but also on the thickness of the shelf, the surface temperature and the rheologic properties of the layer.

## 4 Conclusions

The simple analysis presented here shows how stresses internal to a floating ice shelf can affect the bending of the shelf edge. It shows that Reeh (1968) was right to be concerned about variations of viscosity through an ice sheet, since those stress determine whether the edge if a shelf bends up or down. For small variations of viscosity across an ice shelf (less than about a factor of about 5) the edge bends down while for larger viscosity variations the edge bends up. Assuming a fairly standard ice flow law and a linear temperature gradient through a shelf, these viscosity variations are controlled by two parameters: the
surface temperature, $T_S$, and the rheologic parameter ratio of activation volume divided by the power law exponent, $Q/n$. The constant $A$ in the flow-law relation (Eq. (11)) should affect the time over which bending deflections occur, but not the direction or maximum magnitude of those deflections. This "internal moment" model predicts upward bending for cold surface temperatures and large values of the ratio $Q/n$.

Significant departures of ice shelf temperature-depth profiles from linearity, as seen for parts of some ice shelves (see Fig. 8),
can affect the bending amplitude and sign. As shown in Fig. 9 rapid surface accretion and/or basal melting can diminish the internal bending moment of an ice shelf. It appears that for many parts of the Ross, the Filchner-Ronne and Amery Ice Shelves the temperature gradient is nearly linear so the simple analytic estimates of internal bending moments should be justified. The most strongly non-linear temperature profiles shown in Fig. 9(d) for one site on the Ross Ice Shelf (RIS) may explain the downward bending seen in a few places along the front of that shelf (Becker et al., 2012).

The edge of the Ross Ice shelf is the only place where a systematic study of shelf bending has been done along an entire shelf front and it offers a good first test of this model for several reasons: RIS shows rampart and moat structures along most of the front (Becker et al., 2021); it has a low surface temperature (e.g. MacAyeal and Thomas, 1979); and no benches have been reported there. The last point is significant since shelf-edge benches have been seen for several other ice shelves (e.g. Scambos et al., 2005) including recent studies using ICESat-2 lidar (Philipp Arndt, personal comm., 2023).

Becker et al. (2021) show strong and systematic variations in the height of upward bending (or in the difference in elevation between the moat and rampart, $\Delta e$) and the inferred ice thickness, $h$. They find that for $h = 150\ m$ the average elevation difference, $\Delta e_M = \sim 5$ m, while for $h = 250\ m$ they find that $\Delta e = \sim 10$ m. As noted above the internal moment model predicts

that the elevation difference scales with the ice layer thickness to the power 3/2 (i.e. with $h^{3/2}$), and so is consistent with these observations. In contrast, the bench model elevations are very weakly dependent on layer thickness (i.e. $\Delta e_B \sim h^{1/4}$) and so do not explain this trend.

The horizontal offset between ramparts and moats along the RIS front also may be easier to explain with the internal moment model than with the bench model. This offset is slightly less than the layer thickness as estimated by Becker et al. (2021). As shown in Fig. 2 the lateral scale of deflections for the internal moment model is roughly a factor of two smaller that of the bench model. A detailed comparison of such model predictions must await numerical viscoelastic models since the flexure parameter is not determined in the present analysis.

This analysis makes several significant approximations that can be considered in numerical simulations. Such simulations are now being done (e.g. Glazer and Buck, 2023) and can treat fully two-dimensional deformation, non-uniform vertical temperature gradients, variations of ice density with depth, the time dependence of viscoelastic deformation, among other factors. Of particular importance will be the calculation of the evolution of the effective flexural wavelength. However, the present analysis can guide those model studies since it suggests testable scaling relations including the dependence of deflection amplitude on the rheologic parameter $Q/n$ and the temperature difference across an ice shelf.

Figure 6 shows that for the internal moment model to explain the observed 5-10 meters of upward bending seen along RIS edge requires that the value of the rheologic ratio $Q/n$ be greater than about 50 kJ/mol. This is at the high end of laboratory, theoretical and field estimates of this ratio, though uncertainties in estimates of these parameters are large. For example, estimates for the flow-law exponent, $n$, vary from 1 to 4.5, with higher values corresponding with faster-flowing ice, less sliding, and larger grain sizes (Bons et al., 2018; Zeitz et al., 2020; Millstein et al., 2022, Behn et al., 2021). Likewise, $Q$ is found to vary with temperature (Barnes et al., 1997; Paterson, 1991) with estimated values for cold ice (<-10 °C) ranging from 42 to 85 kJ/ mol; and for warmer ice from 120 to 200 kJ/mol (Zeitz et al., 2020; Greve, 1997; Furst et al., 2011; Lipscomb et al., 2019; Weertman et al., 1983). Another key result of laboratory studies of ice flow is that at low stress differences ($\sim <10^5$ Pa) flow is better described by low values of the power-law exponent $n$ (e.g. Behn et al., 2021). An ice shelf with a cold surface temperature the warm lower part of the shelf may only support stress differences on the order of $10^4$ Pa and so may be in a low n flow regime (as can be seen by combining an estimate of average stress difference via Eq. (7) with the calculation of vertical variations in effective viscosity given by Eq. (12), and illustrated in Fig. 3).

Further numerical and observational tests of the internal moment model may allow new constraints to be places on these rheologic parameters and since models of ice sheet and ice shelf flow depends on these parameters there should be interest in doing such tests. One important observation is to measure the distribution and size of ice benches since they certainly can affect shelf bending (e.g. Wagner et al., 2014; 2016; Sartore et al., 2024). Key observational tests should also involve comprehensive studies of shelf-edge bending for all the ice shelves of Antarctica and Greenland. Surface temperatures vary

for different ice shelves and if the banding characteristics vary systematically with surface temperature this offers hope of giving new constraints on the temperature dependence of ice rheology.

**Code and data availability: Neither numerical codes nor new data sets were used to prepare this contribution.**

**Author contribution:** All work from conceptualization to writing was done by the sole author.

**Competing interests**: **The author declares that he has no conflict of interest.**

**Acknowledgements.** Thanks to Emily Glazer, Ching-Yao Lai, Till Wagner, Nicolas Sartore, Phillip Arndt, Niall Coffey, Kirsty Tinto and Andrew Hoffman for helpful discussions and comments. This research did not receive any specific grant from funding agencies in the public, commercial, or not-for-profit sectors.

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
