# Peer review of "The effect of ice shelf rheology on shelf edge bending"

_EGUsphere, 2024_

## Author Comment (AC1)

I cannot remember when I have had more helpful reviews than those given by the two people who carefully went through my paper. The fact that they both noted some of the same issues made it certain that I had to fix a few things. I describe the planned changes in detail after each of the related reviewer comments which are italicized.

I did have to abandon my effort to make the paper totally analytic in that I had to do numerical integrations to show the effect of non-linear temperature profiles. Since this took considerable effort, I may not have addressed all the other comments as well as I should. The most important changes are:

1) To try to make the model idea better I plan to add a description of viscous bending to the introduction and in section 3.3 on the thin plate approximation of the flexural wavelength.

2) To illustrate the asymptotic behavior of the analytic model I added a panel to Figure 5 showing predicted internal bending moments versus the log of the e-folding length for viscosity variations.

3) I analyzed the errors in the analytic approximation by carrying out numerical integration of the stress differences for the assumed ice flow law. The difference between the full and approximate solutions for the internal bending moment depend on the assumed flow law parameters and surface temperature, but are less than 3% for the most extreme cases illustrated in the figures.

4) A new section on "Effects of nonlinear temperature variation with depth" will include (1) a figure showing temperatures from 3 boreholes on the Ross Ice Shelf and 1 from the Amery Ice shelf, and (2) calculation of steady-state temperature profiles for a range of rates of surface accretion and basal accretion or melting (after Robin, 1955) and results of numerical analysis of internal bending moments for those temperature profiles. The Figures are given below.

*Reviewer1*

*This manuscript presents a novel analysis of flexure at the terminus of a freely floating ice shelf. It addresses observations of upward flexure of the Ross Ice shelf near Roosevelt island.*

I plan to note that the upward flexure is seen not only near Roosevelt Island but along along~74% of the front of Ross Ice Shelf (Becker et al., 2021).

*Whereas these were previously explained in terms of an eroded ice bench, this manuscript shows that a vertical variation in ice viscosity, arising from a linear variation in ice temperature and acting on a vertically uniform rate of extension, gives rise to a bending moment that can explain the observed flexure. This is an appealing hypothesis because ice benches are not observed at Ross (using the same sensor that has observed them elsewhere). The argument is supported by a clear and relatively simple mathematical theory that is consistent with a classical analysis of ice shelves (Weertman 1957). The manuscript is well written and well illustrated, easy to follow, and makes a novel and significant contribution to our understanding of ice-shelf dynamics. It has important implications for our understanding of ice-shelf calving. It should be published with minor revisions.*

*I see no major problems with the manuscript as written. The author has cleverly applied insights from plate flexure derived in the context of tectonics to ice shelves. Amusingly, the author*

*highlights that his analysis was tee'd up by Reeh 1968, whose "mathematical troubles" are relieved by the simplifying assumption of a linear temperature variation through the ice shelf. This leads to a Taylor series expansion and truncation at leading order, making the moment integral tractable and unlocking a solution. This context prompts two relatively minor suggestions. The first is to better discuss and justify the linear temperature assumption, as this is crucial to progress. There are borehole measurements by Mike Craven et al (e.g., J. Glaciology, Vol. 55, No. 192, 2009) and likely others. Plotting their data in comparison to a linear fit might be nice.*

Here is a figure and caption I plan to add to an added section 3.4 Effects of nonlinear temperature variation with depth:

[Figure]

New Figure 9. 4 sets of borehole temperature measurements that constrain the temperature profiles for parts of two Antarctic ice shelves. (a, b and c) are for the Ross ice shelf re-plotted from Taylor and MacAyeal (1979)

while (d) is from the Amery Ice Shelf re-plotted from Craven et al. (2009) and locations of the boreholes are given in those references.

In that new section I propose to discuss the effect of "Robin-type" temperature profiles on the calculated internal moments. To do that I also had to use a numerical integration of the stress differences predicted by the full flow law using the non-linear temperature profiles. I would add something like this:

Several effects, including accretion or melting of the surface or base of an ice shelf, can contribute to non-linearity of temperatures with depth and this will affect stresses and so internal moments. Following Robin (1955) we can estimate the effects of surface accretion and basal accretion or melting on temperatures in an ice shelf ice shelf. Pure shear thinning of the layer maintains a uniform shelf thickness, $h$, while the velocity of the surface is $v_S$ and the velocity of the vase is $v_B$. The surface is maintained at $T_S$ and the base at $T_B$. The steady-state temperature $T$ as a function of depth below the ice surface, $z$, can be written:

$$T(z^*) = T_S + (T_B - T_S)\left\{\frac{\text{erf}(\xi z^*) - \text{erf}(-\xi z_{ref})}{\text{erf}[\xi(1-z_{ref})] - \text{erf}(-\xi z_{ref})}\right\}$$

(to be added as equation 23)

where $z^* = {}^z/_h - z_{ref}$, $z_{ref} = \frac{v_S}{(v_S-v_B)}$, $\xi = \sqrt{\frac{(v_S-v_B)h}{2\kappa}}$ and $\kappa$ is the thermal diffusivity, here taken to be $10^{-6}$ m$^2$/s. Resulting temperature profiles for a layer 400 m thick and several combinations of $v_S$ and $v_B$ are shown in the figure below (proposed new Figure 10). Equation (11) was then used with the calculated temperatures with depth to give the stress differences needed for numerical integration of equations 5 and 10 to get the predicted internal moment of the horizontal stress distribution (M$_I$).

[Figure]

[Figure]

New Figure 10.  Effect of surface or basal accretion on ice shelf temperature profiles and internal bending moments. (a) Examples of three steady-state temperature profiles for the indicated values of surface and basal velocities ($v_S$ and $v_B$) compared with a linear temperature profile for the same surface and base temperatures. (b) Numerically calculated normalized internal bending moments for a range of temperature profiles calculated with the indicated parameters. Black dots indicate the temperature profiles shown in part (a). Moments are divided by the moment for a linear temperature profile.

In describing this in the text I plan to note that freezing onto the base of an ice shelf should act to increase the amplitude of the internal moment and so the expected upward bending of the shelf edge.  Accretion to the surface and melting of the base act to diminish the internal moment.

*The second is to use the second-order term in the Taylor series as a means to estimate the truncation error in equation (14).  My quick calculation gives a multiplicative factor of exp[( T'/T_s z )²].  Taking z=h as an upper bound, this gives exp( \delta T/T_s)² ) ~ exp( (30/240)² ) ~ 1.02.  So a maximum 2% error in viscosity due to Taylor expansion.  This could be propagated through the calculation to obtain the error on M_I (but in fact the linear temperature assumption must be a larger source of error).*

Rather than do that I used the full assumed flow law (Equation 11 in my text) and numerically integrate the resulting stress differences based on equation (12) to calculate the internal moment using equation (10). As noted above, the difference between the full and approximate solutions for the internal bending moment depend on the assumed flow law parameters and surface temperature, but are less than 3% for the most extreme cases illustrated in the figures.  In going through this exercise, I realized that I had not updated the last version of the equation that I used to calculate the e-folding length for viscosity variations in equation (15).  So now equation (15) becomes:

$$z_0 = \frac{nRT_ST_B}{Q^{dT}/_{dz}} \quad \text{or} \quad \frac{z_0}{h} = \frac{nRT_ST_B}{Q(T_B - T_s)} \tag{15}$$

 Changing the "$Ts^2$" term in that equation with "$TsTb$" forces the log-linear approximation to pass equal the viscosity or stress difference values at the top and base of the layer as given by the full flow law.  This significantly reduces the errors of the moment calculated analytically.  I had done that in the cases illustrated in the submitted text (as can be seen in Figure 4(b)) but had not updated equation (15).

*My third suggestion is to more carefully discuss the time-dependence of viscoelastic flexure.  Although the details will vary between problems, the scaling with time/(Maxwell time) should not.  How does this affect the comparison with the Ross ice shelf?  What is the age of that edge?  Is it fresh (i.e., age/Maxwell << 1)?  This relates to the approximate of stresses as, close to the shelf edge, they will be modified with time since calving.  In this regard, the bi-metallic strip analogy is somewhat misleading, as it is in mechanical equilibrium at a fixed temperature.*

In response to this suggestion and a similar one made by reviewer 2 I plan to add a paragraph to sections 2 "Conceptual Model" describing viscous effects.

Also, in explaining viscous bending effects more thoroughly in section 3.3 on "Topographic variation…" I will replace the paragraph beginning on line 262 with:

Reeh (1968) and Olive et al. (2016) find that for a viscous or viscoelastic plate with a uniform viscosity $\eta$ the wavelength of the flexure changes with time as:

$$\alpha(t)/\alpha(t=0) \sim \left[\frac{\tau_M}{t}\right]^{1/4} \text{ and}$$

$$\alpha(t=0) \sim \left[\frac{Eh^3}{\rho_w g(1-v^2)}\right]^{1/4}. \tag{21}$$

where $t$ is time, E is Young's Modulus and $v$ is Poisson's ratio and where $\tau_M = \frac{Young's Modulus}{average\ viscosity}$ is a measure of the Maxwell time of the layer. Combining equations 20 and 21 suggests that the amplitude should increase with time roughly as: $e_0^M(t) = e_0^M(0) \left[\frac{t}{\tau_M}\right]^{1/2}$. Eventually, as the flexure parameter approaches the layer thickness, the two-dimensional nature of the problem means that the thin-plate approximation is no longer valid. For layer of a few hundred meters thickness this should take about 1000 Maxwell times. For an average layer viscosity of a few times $10^{14}$ Pas and a layer thickness of 300 m this should take about 4 years. Reeh (1968) came to this conclusion and estimated that the long-term flexure parameter can be a bit smaller than the layer thickness. A more thorough study was done by Olive et al. (2016) who compare fully two-dimensional viscoelastic models of flexure to the thin plate solution and find the best fitting thin plate flexure parameter evolution to match the 2D results. They find that after many Maxwell times that the effective flexure parameter is smaller than the layer thickness. Thus, for figures 6 and 7 $\alpha$ is set to 250 m while the ice layer thickness is taken to be 400 m.

Finally, I plan to add a sentence or two to section 4 "Discussion and Conclusions" describing how IceSat II lidar observations analyzed by Sartore et al., (2024) show that after a calving event on part of the front of the Ross Ice Shelf that the moat and rampart takes several years to develop and grow. This cold be explained in terms of the expected viscous change in the flexural wavelength and the resulting increase in bending deflections.

*Broadly, I think the author should draw more attention to the assumptions made and the caveats and cautions that they introduce. This would not detract from the importance of the manuscript, but would better promote further research to build and test the ideas introduced here.*

I tred to do this mainly with the new section on temperature profiles describe above.

*Some detailed points, by line number in the manuscript:*

*[32] where ice shelf serves as an adjective, it should have a hyphen. E.g., ice-shelf edges*

Will change!

*[Fig 1] expand the figure caption to explain the lines in these figures. Improve the resolution to clarify that the hashing are ascending and descending track lines.*

It is worth noting that Reviewer 2 also wanted a better explanation. I plan to add this to the figure caption: "The grey lines are estimated streamlines of ice flow while the red lines show both ascending and descending IceSat II track lines analyzed by Becker et al. (2021)."

*[76--78] These two sentences say the same thing, which is confusing. Only one is needed.*

I will delete the second sentence.

*[98--99] The sentence starting with "Imagine" is important but the reader hasn't yet been adequately informed about why. Somewhere above (maybe the introduction) there should be a brief discussion of how visco-elastic bending is time dependent.*

As noted above I will add a couple of sentences to the conceptual model and a revised paragraph in section 3.3.

[103] "To do this" grammatical issue here.

I plan to find a proper grammarian who can tell me what is wrong here.

*[163] The result here appear to be positive but represents downward flexure (line 124 states that upward bending corresponds to positive total applied moments). Please check signs.*

Correct, I left out the minus sign and will put it in. I also have to add a minus sign to equation (19).

*[175] Spelling of MacAyeal.*

Will fix.

*[188] The assumption regarding stresses evaluated at large distance from the edge of the ice is somewhat sketchy so I think a bit more emphasis and discussion would be relevant here.*

I will add reference to Weertman (1957) on this topic as suggested by reviewer 2.

*[210] A reference here to Weertman 1957 or similar would be appropriate and helpful.*

I will add this as well.

*[Fig 5b] I think that a version of this plot with a logarithmic x axis (and an expanded domain and range) would be helpful in seeing the asymptotic behaviour of M_I at large and small z_0/h.*

Good idea. In the text I will note that the analytic internal moment solution for small values of $z_0$ goes to -3.75 times the value of the water related moment. For large values of $z_0$ it goes to zero. Here is the new version of Figure (5) with an added plot (c):

[Figure]

New Figure 5c (to be added to present Figure 5) shows the variation of internal and total moments as functions of the logarithm of the e-folding scale if viscosity variations.

*[294] "illustrates shows"*

Will cut one.

*[340] "places"*

Will be "placed"

*[throughout] mathematical notation should be italic but frequently appears as regular next.*

Will change.

Added references:

Craven, M., Allison, I., Fricker, H.A., Warner, R.: Properties of a marine ice layer under the Amery Ice Shelf, Antarctica, J. Glaciol. 2009; 55: 717-728. doi:10.3189/00221430978947094

Sartore, N. B., Wagner, T. J. W., Siegfried, M. R., Pujara, N., and Zoet, L. K.: Calving of Ross Ice Shelf from wave erosion and hydrostatic stresses, EGUsphere [preprint], https://doi.org/10.5194/egusphere-2024-571, 2024.

Thomas, R. H. and MacAyeal, D. R.: Derived cahracteristics of the Ross Ice Shelf, Antarctica, J. Glaciol. 1982; 28:397-412. doi:10.3189/S0022143000005025

---

## Author Comment (AC2)

I cannot remember when I have had more helpful reviews than those given by the two people who carefully went through my paper. The fact that they both noted some of the same issues made it certain that I had to fix a few things. I describe the planned changes in detail after each of the related reviewer comments which are italicized.

I did have to abandon my effort to make the paper totally analytic in that I had to do numerical integrations to show the effect of non-linear temperature profiles. Since this took considerable effort, I may not have addressed all the other comments as well as I should. The most important changes are:

1) To try to make the model idea better I plan to add a description of viscous bending to the introduction and in section 3.3 on the thin plate approximation of the flexural wavelength.

2) To illustrate the asymptotic behavior of the analytic model I added a panel to Figure 5 showing predicted internal bending moments versus the log of the e-folding length for viscosity variations.

3) I analyzed the errors in the analytic approximation by carrying out numerical integration of the stress differences for the assumed ice flow law. The difference between the full and approximate solutions for the internal bending moment depend on the assumed flow law parameters and surface temperature, but are less than 3% for the most extreme cases illustrated in the figures.

4) A new section on "Effects of nonlinear temperature variation with depth" will include (1) a figure showing temperatures from 3 boreholes on the Ross Ice Shelf and 1 from the Amery Ice shelf, and (2) calculation of steady-state temperature profiles for a range of rates of surface accretion and basal accretion or melting (after Robin, 1955) and results of numerical analysis of internal bending moments for those temperature profiles. The Figures are given below.

*Reviewer 2*

*Summary:*

*The manuscript extends the theory first developed in Reeh. 1968 and convincingly proposes a new theory that can explain upward ice shelf bending without requiring the introduction of geometric features that aren't always observed, as required the "bench" model. The paper is well-written and impactful while managing to remain brief. I do think some improvement in clarity could be offered by increasing the level of detail for some conceptual ideas. The paper, overall, is excellent and warrants prompt publication.*

*Suggestions:*

1. *In the conceptual model section, bending generated by vertical variation in horizontal stress and associated internal moments is discussed. However, I am a little confused about the direction of the bending moment. If "there is a decrease in extensional stress with depth," would this not correspond to a top out sense of bending? I am sure I am misunderstanding this in a perhaps amateurish way, but I am also sure that other readers will have this same confusion. I think the manuscript would benefit from a bit more detail explaining, from a*

*conceptual perspective, the direction of internal moments generated by horizontal stresses that decrease in magnitude with depth, and the direction of the equivalent edge moment. Right now, it feels like clarity is sacrificed a bit in exchange for brevity.*

I had to think about this for a while and this is what I have come up with so far:

"The effect of internal moments in layer with zero external moments can is equivalent to applying external moments of the opposite sign and same magnitude as the internal moments to a layer with zero internal moments. The problem of bending boundary conditions is analogous to the simple problem of uniaxial stress. Consider a finite length elastic bar with stress free sides. One end of the bar is fixed and the other end can be pushed or pulled. If the bar is initially unstressed and then pushed with a horizontal compressive force $F_C$ the end will be pushed in a distance $\Delta x$ and the internal horizontal stress will be uniformly compressive. The deformation is the same if the layer was first pulled with an extensional force so that the initial internal force $F_E$ = - $F_C$. Setting the boundary force to zero would result in compression of the layer and movement of the layer by a distance $\Delta x$. So the deformation is the same if a force opposite to the initial internal force is applied to the and unstressed bar."

*Detailed comments:*

*[1] I feel that the introduction could flow slightly better if the first paragraph started broad and then narrowed (i.e. ice shelf breakup is important because it can enhance sea level rise -> one of the processes that cause breakup is bending -> bending causes breakup in the following two ways). This is just personal preference and does not need to be addressed unless the author agrees.*

Good idea. I will do that in the revised introduction.

*[40-41] Could be a good idea to mention a few ice shelves that show bending consistent with the typical downward bending and which ice shelves show the opposite sense of bending.*

I will do this and describe observations in Scambos et al. (2005).

*[51-57] This is written with panache and is fun to read!*

I was surprised to find the line in Reeh (1968) buried in his model description and neither in the introduction nor the conclusions.

*[65] Presumably the gray lines are flow lines, but it might be good to mention this briefly in the caption.*

As noted in the response to reviewer 1 I now describe the figure more completely and note that the grey lines are flow lines (see above).

*[68] For context, it could be good to say something like "Ice shelves that are not heavily buttressed are under extension (Weertman, 1957). While ice shelves are typically assumed to*

*have negligible vertical gradients in horizontal stress, significant vertical variation in viscosity generates* vertical gradients in horizontal stress that cannot be neglected."

Good suggestion. I will make that change.

*[74] After this, we look at a couple examples in other geophysical settings. It could be good to end this paragraph, or start the next, with a sentence like "Similar physics are observed in several geophysical settings." Right now, the jump is a little abrupt feeling.*

I'll add sentences noting the origin of the internal stresses in these cases, saying something like: The internal stresses in the Haxby and Parmentier (1989) model consider an increase in extensional stresses with depth arising from thermal contraction combined with shallow yielding. In the Buck (2001) model the lithosphere is accreted by addition of magmatic dikes that widen with depth in response to ridge axial stresses. This build lithosphere with intrinsic curvature that flattens with distance from the spreading axis. The flattening results in surface extensional faulting consistent with the observed growth of fault offset with distance from the axis."

*[75-80] This paragraph is a little confusing. I would explicitly describe what causes vertical variation in horizontal stress in the lithospheric case. "Those authors note that gravity prevents lithospheric bending that is of much longer wavelength then the effective flexural wavelength of the layer. At very long length scales gravity prevents the layer from bending." is redundant. Is gravity generating the vertical variation in horizontal stress? I would just explain this analogous case with a bit more detail to improve clarity.*

Yes, reviewer 1 also noted the redundancy and I'll cut the 2nd sentence.

*[81] Remove "that that."*

Yes yes.

*[84] Should be "is equivalent" not "are equivalent."*

Will change

*[84-85] I understand the general concept that uniform internal loading is equivalent to remote loading (common in fracture mechanics, for instance, when considering stresses acting on the edges of a cracked plate and stresses acting on the interior crack walls). It is not entirely clear to me why the equivalent on the end of a layer has opposite direction, however. A bit more explanation could be helpful. This ties into the broader comment above.*

See my response to "Suggestion 1" above.

*[110] This may just be how the draft version is typeset, but this figure is a little confusing to see before the reference horizontal stress has actually been defined.*

I will move the figure to be after the definition of the reference stress.

*[143] "water pressure" not "pressures"*

Will change

*[196] Quick half sentence explaining e-folding could be good. Most readers will probably already know, but clarity is always to be encouraged, especially because z_0 is an important parameter moving forward.*

Yes, I'll add something like: "(i.e. the distance over which the viscosity drops by 1/e)"

*[340] "may allow new constraints to be placed" not "places."*

It will be so placed.

Added references:

Craven, M., Allison, I., Fricker, H.A., Warner, R.: Properties of a marine ice layer under the Amery Ice Shelf, Antarctica, J. Glaciol. 2009; 55: 717-728. doi:10.3189/002214309789047094

Sartore, N. B., Wagner, T. J. W., Siegfried, M. R., Pujara, N., and Zoet, L. K.: Calving of Ross Ice Shelf from wave erosion and hydrostatic stresses, EGUsphere [preprint], https://doi.org/10.5194/egusphere-2024-571, 2024.

Thomas, R. H. and MacAyeal, D. R.: Derived cahracteristics of the Ross Ice Shelf, Antarctica, J. Glaciol. 1982; 28:397-412. doi:10.3189/S0022143000005025

---

## Author Comment (AC3)

To replace Figure 3 (b) and (c)

---

## Author Response (AR1)

1)  To try to make the model idea better I added a description of viscous bending to the introduction and in section 3.3 on the thin plate approximation of the flexural wavelength.

2) I added an example of a downward bending shelf-edge profile to figure 1 and changed the map.

3) To illustrate the asymptotic behavior of the analytic model I added a panel to Figure 5 showing predicted internal bending moments versus the log of the e-folding length for viscosity variations.

4) I analyzed the errors in the analytic approximation by carrying out numerical integration of the stress differences for the assumed ice flow law.  The difference between the full and approximate solutions for the internal bending moment depend on the assumed flow law parameters and surface temperature, but are less than 3% for the most extreme cases illustrated in the figures and this is noted in the text.

5)  A new section on "Effects of nonlinear temperature-depth profiles on shelf bending"  include an added figure 8 showing temperatures from 3 boreholes on the Ross Ice Shelf, 1 from Filchner-Ronne and 1 from the Amery Ice shelf, and a new added figure 9 showing results of calculation of steady-state temperature profiles for a range of rates of surface accretion and basal accretion or melting (after Robin, 1955) and results of numerical analysis of internal bending moments for those temperature profiles.

Here are specific responses to the reviewers:

*Reviewer1*

*This manuscript presents a novel analysis of flexure at the terminus of a freely floating ice shelf.  It addresses observations of upward flexure of the Ross Ice shelf near Roosevelt island.*

I plan to note that the upward flexure is seen not only near Roosevelt Island but along along~74% of the front of Ross Ice Shelf (Becker et al., 2021).

*Whereas these were previously explained in terms of an eroded ice bench, this manuscript shows that a vertical variation in ice viscosity, arising from a linear variation in ice temperature and acting on a vertically uniform rate of extension, gives rise to a bending moment that can explain the observed flexure.  This is an appealing hypothesis because ice benches are not observed at Ross (using the same sensor that has observed them elsewhere).  The argument is supported by a clear and relatively simple mathematical theory that is consistent with a classical analysis of ice shelves (Weertman 1957).  The manuscript is well written and well illustrated, easy to follow, and makes a novel and significant contribution to our understanding of ice-shelf dynamics.  It has important implications for our understanding of ice-shelf calving.  It should be published with minor revisions.*

*I see no major problems with the manuscript as written.  The author has cleverly applied insights from plate flexure derived in the context of tectonics to ice shelves.  Amusingly, the author highlights that his analysis was tee'd up by Reeh 1968, whose "mathematical troubles" are*

*relieved by the simplifying assumption of a linear temperature variation through the ice shelf. This leads to a Taylor series expansion and truncation at leading order, making the moment integral tractable and unlocking a solution. This context prompts two relatively minor suggestions. The first is to better discuss and justify the linear temperature assumption, as this is crucial to progress. There are borehole measurements by Mike Craven et al (e.g., J. Glaciology, Vol. 55, No. 192, 2009) and likely others. Plotting their data in comparison to a linear fit might be nice.*

See note (5) above.

*The second is to use the second-order term in the Taylor series as a means to estimate the truncation error in equation (14). My quick calculation gives a multiplicative factor of exp[( T'/T_s z )²]. Taking z=h as an upper bound, this gives exp( (\delta T/T_s)² ) ~ exp( (30/240)² ) ~ 1.02. So a maximum 2% error in viscosity due to Taylor expansion. This could be propagated through the calculation to obtain the error on M_I (but in fact the linear temperature assumption must be a larger source of error).*

Rather than do that I used the full assumed flow law (Equation 11 in my text) and numerically integrate the resulting stress differences based on equation (12) to calculate the internal moment using equation (10). As noted above, the difference between the full and approximate solutions for the internal bending moment depend on the assumed flow law parameters and surface temperature, but are less than 3% for the most extreme cases illustrated in the figures. In going through this exercise, I realized that I had not updated the last version of the equation that I used to calculate the e-folding length for viscosity variations in equation (15). So now equation (15) becomes:

$$z_0 = \frac{nRT_ST_B}{Q \, {dT}/{dz}} \quad \text{or} \quad \frac{z_0}{h} = \frac{nRT_ST_B}{Q(T_B - T_s)} \tag{15}$$

Changing the "$T_s^2$" term in that equation with "$T_sT_b$" forces the log-linear approximation to pass equal the viscosity or stress difference values at the top and base of the layer as given by the full flow law. This significantly reduces the errors of the moment calculated analytically. I had done that in the cases illustrated in the submitted text (as can be seen in Figure 4(b)) but had not updated equation (15).

*My third suggestion is to more carefully discuss the time-dependence of viscoelastic flexure. Although the details will vary between problems, the scaling with time/(Maxwell time) should not. How does this affect the comparison with the Ross ice shelf? What is the age of that edge? Is it fresh (i.e., age/Maxwell << 1)? This relates to the approximate of stresses as, close to the shelf edge, they will be modified with time since calving. In this regard, the bi-metallic strip analogy is somewhat misleading, as it is in mechanical equilibrium at a fixed temperature.*

See note (1) above.

*Broadly, I think the author should draw more attention to the assumptions made and the caveats and cautions that they introduce. This would not detract from the importance of the manuscript, but would better promote further research to build and test the ideas introduced here.*

I tried to do this mainly with the new section on temperature profiles describe above.

*Some detailed points, by line number in the manuscript:*

*[32] where ice shelf serves as an adjective, it should have a hyphen. E.g., ice-shelf edges*

Changed

*[Fig 1] expand the figure caption to explain the lines in these figures. Improve the resolution to clarify that the hashing are ascending and descending track lines.*

It is worth noting that Reviewer 2 also wanted a better explanation. I plan to add this to the figure caption: "The grey lines are estimated streamlines of ice flow while the red lines show both ascending and descending IceSat II track lines analyzed by Becker et al. (2021)."

*[76--78] These two sentences say the same thing, which is confusing. Only one is needed.*

I deleted the second sentence.

*[98--99] The sentence starting with "Imagine" is important but the reader hasn't yet been adequately informed about why. Somewhere above (maybe the introduction) there should be a brief discussion of how visco-elastic bending is time dependent.*

As noted above I added a couple of sentences to the conceptual model and a revised paragraph in section 3.3.

*[103] "To do this" grammatical issue here.*

Changed

*[163] The result here appear to be positive but represents downward flexure (line 124 states that upward bending corresponds to positive total applied moments). Please check signs.*

Correct, I left out the minus sign and now put it in. I also have added a minus sign to equation (19).

*[175] Spelling of MacAyeal.*

Corrected.

*[188] The assumption regarding stresses evaluated at large distance from the edge of the ice is somewhat sketchy so I think a bit more emphasis and discussion would be relevant here.*

I added reference to Weertman (1957) on this topic as suggested by reviewer 2.

*[210] A reference here to Weertman 1957 or similar would be appropriate and helpful.*

Added this as well.

*[Fig 5b] I think that a version of this plot with a logarithmic x axis (and an expanded domain and range) would be helpful in seeing the asymptotic behaviour of $M\_I$ at large and small $z\_0/h$.*

Done.

Cut one.

*[340] "places"*

Will be "placed"

*[throughout] mathematical notation should be italic but frequently appears as regular next.*

Changed.

*Reviewer 2*

*Suggestions:*

1. *In the conceptual model section, bending generated by vertical variation in horizontal stress and associated internal moments is discussed. However, I am a little confused about the direction of the bending moment. If "there is a decrease in extensional stress with depth," would this not correspond to a top out sense of bending? I am sure I am misunderstanding this in a perhaps amateurish way, but I am also sure that other readers will have this same confusion. I think the manuscript would benefit from a bit more detail explaining, from a conceptual perspective, the direction of internal moments generated by horizontal stresses that decrease in magnitude with depth, and the direction of the equivalent edge moment. Right now, it feels like clarity is sacrificed a bit in exchange for brevity.*

I changed some of the description of the conceptual models and altered the illustration of the idea in Figure 3

*Detailed comments:*

*[1] I feel that the introduction could flow slightly better if the first paragraph started broad and then narrowed (i.e. ice shelf breakup is important because it can enhance sea level rise -> one of the processes that cause breakup is bending -> bending causes breakup in the following two ways). This is just personal preference and does not need to be addressed unless the author agrees.*

Good idea. I revised the introduction.

*[40-41] Could be a good idea to mention a few ice shelves that show bending consistent with the typical downward bending and which ice shelves show the opposite sense of bending.*

I did this for Figure 1 and described observations in Scambos et al. (2005).

*[51-57] This is written with panache and is fun to read!*

*[65] Presumably the gray lines are flow lines, but it might be good to mention this briefly in the caption.*

Changes the map in figure 1

*[68] For context, it could be good to say something like "Ice shelves that are not heavily buttressed are under extension (Weertman, 1957). While ice shelves are typically assumed to have negligible vertical gradients in horizontal stress, significant vertical variation in viscosity generates vertical gradients in horizontal stress that cannot be neglected."*

Good suggestion. I made that change.

*[74] After this, we look at a couple examples in other geophysical settings. It could be good to end this paragraph, or start the next, with a sentence like "Similar physics are observed in several geophysical settings." Right now, the jump is a little abrupt feeling.*

I added sentences noting the origin of the internal stresses in these cases, saying something like: The internal stresses in the Haxby and Parmentier (1989) model.

*[75-80] This paragraph is a little confusing. I would explicitly describe what causes vertical variation in horizontal stress in the lithospheric case. "Those authors note that gravity prevents lithospheric bending that is of much longer wavelength then the effective flexural wavelength of the layer. At very long length scales gravity prevents the layer from bending." is redundant. Is gravity generating the vertical variation in horizontal stress? I would just explain this analogous case with a bit more detail to improve clarity.*

Yes, reviewer 1 also noted the redundancy and I cut the 2nd sentence.

*[81] Remove "that that."*

Done.

*[84] Should be "is equivalent" not "are equivalent."*

Changed

*[84-85] I understand the general concept that uniform internal loading is equivalent to remote loading (common in fracture mechanics, for instance, when considering stresses acting on the edges of a cracked plate and stresses acting on the interior crack walls). It is not entirely clear to me why the equivalent on the end of a layer has opposite direction, however. A bit more explanation could be helpful. This ties into the broader comment above.*

See my response to "Suggestion 1" above.

*[110] This may just be how the draft version is typeset, but this figure is a little confusing to see before the reference horizontal stress has actually been defined.*

I moved the figure to be after the definition of the reference stress.

*[143] "water pressure" not "pressures"*

Changed

*[196] Quick half sentence explaining e-folding could be good. Most readers will probably already know, but clarity is always to be encouraged, especially because z_0 is an important parameter moving forward.*

Yes, I added: "(i.e. the distance over which the viscosity drops by 1/e)"

*[340] "may allow new constraints to be placed" not "places."*

It was so placed.

Added references:

Craven, M., Allison, I., Fricker, H.A., Warner, R.: Properties of a marine ice layer under the Amery Ice Shelf, Antarctica, J. Glaciol. 2009; 55: 717-728. doi:10.3189/002214309788947094

Sartore, N. B., Wagner, T. J. W., Siegfried, M. R., Pujara, N., and Zoet, L. K.: Calving of Ross Ice Shelf from wave erosion and hydrostatic stresses, EGUsphere [preprint], https://doi.org/10.5194/egusphere-2024-571, 2024.

Thomas, R. H. and MacAyeal, D. R.: Derived cahracteristics of the Ross Ice Shelf, Antarctica, J. Glaciol. 1982; 28:397-412. doi:10.3189/S0022143000005025